# ON THE ROLE OF DIFFICULT PROMPTS IN SELF-PLAY PREFERENCE OPTIMIZATION

## ABSTRACT

Self-play preference optimization has emerged as a prominent paradigm for aligning large language models (LLMs). It typically involves a language model to generate on-policy responses for prompts and a reward model (RM) to guide the selection of chosen and rejected responses, which can be further trained with direct preference optimization (DPO). However, the role of prompts remains underexplored, despite being a core component in this pipeline. In this work, we investigate how prompts of varying difficulty influence self-play preference optimization. We first use the mean reward of $N$ sampled responses of a prompt as a proxy for its difficulty. We find that difficult prompts exhibit substantially inferior self-play optimization performance in comparison to easy prompts for language models. Moreover, incorporating difficult prompts into training fails to enhance overall performance and, in fact, leads to slight degradation compared to training on easy prompts alone. We also observe that the performance gap between difficult and easy prompts closes as the model capacity increases, suggesting that difficulty interacts with the model capacity. Building on these findings, we explore strategies to mitigate the negative effect of difficult prompts on final performance. We demonstrate that selectively removing an appropriate portion of challenging prompts enhances overall self-play performance, while also reporting failed attempts and lessons learned.

## 1 INTRODUCTION

Large language models (LLMs) have achieved remarkable success in a wide range of natural language processing tasks, but aligning them with human values and preferences remains a challenge (Brown et al., 2020; Wei et al., 2022; Bai et al., 2022; Weidinger et al., 2022; Bubeck et al., 2023; Grattafiori et al., 2024; Ji et al., 2025). Reinforcement learning from human feedback (RLHF) has become a popular approach to align LLMs with human preferences (Stiennon et al., 2020; Ethayarajh et al., 2022; Ouyang et al., 2022; Tang et al., 2024; Qi et al., 2025a). It involves first training a reward model (Gao et al., 2023), which then provides feedback signals to optimize a policy model through reinforcement learning, typically using proximal policy optimization (PPO). To further simplify the procedure, Rafailov et al. (2023) introduced Direct Preference Optimization (DPO), which bypasses the need for reward models when optimizing the policy (Gheshlaghi Azar et al., 2024; Ethayarajh et al., 2024; Meng et al., 2024). However, these methods still heavily rely on manually curated pairwise preference data to effectively optimize policy models (Ethayarajh et al., 2022; Cui et al., 2024; Wang et al., 2024d;c; Raghavendra et al., 2025).

Recently, DPO-based self-play preference optimization has emerged to further enhance the alignment performance of LLMs, which employs standard heuristics or off-the-shelf reward models to select chosen and rejected responses to questions [1] without manual efforts (Tunstall et al., 2024; Song et al., 2024; Chen et al., 2024; Pang et al., 2024; Wu et al., 2025; Li & Khashabi, 2025). For example, $n$ response candidates can be sampled from policy models and scored with a reward model to construct preference pairs for DPO. Specifically, the sample with the highest reward is usually selected as the chosen response, while the one with the lowest reward is selected as the rejected response (Meng et al., 2024; Li & Khashabi, 2025). Previous work has primarily investigated how

---

[1]Prompt and question are exchangeable in this paper.

reward models and the construction of training pairs contribute to preference optimization and overall alignment performance (Tajwar et al., 2024; Gao et al., 2025; Xiao et al., 2025; Qi et al., 2025b), while the role of prompts has often been overlooked despite being a core component of the pipeline.

In this paper, we fill this gap by focusing on the role of prompts in self-play preference optimization pipeline. Given an LLM, we first propose using the mean reward of $N$ sampled responses for a prompt as a proxy of its difficulty. Intuitively, a lower mean reward indicates a higher difficulty for the corresponding prompt. We can sort prompts by their mean reward to obtain a difficulty ranking, then partition them into subsets of varying difficulty. Following that, we find that the quartile of prompts with the lowest mean reward (highest difficulty) yields inferior self-play performance than an equal number of prompts in the remaining easier subset. Furthermore, this quartile of prompts will not lead to performance gains when incorporated into training with the remaining easier prompts. We then introduce three approaches to improve the performance of models by mitigating the hard prompt issue. We find that removing a portion of difficult prompts appropriately will lead to performance gains. In summary, we have following contributions in this work.

First, we use the mean reward of $N$ sampled responses of a prompt as a measure of its difficulty. Our intuition is that prompts with lower mean rewards are considered more difficult than those with higher mean rewards, allowing us to sort prompts by difficulty. We find that 10 samples per prompt suffice to obtain a stable difficulty ranking of the questions. In addition, we demonstrate that the difficulty of prompts for an LLM can transfer to another LLM to some extent. Furthermore, we observe that two reward models trained with different loss design and training datasets have similar difficulty assessments, reinforcing the robustness of our metric (**Section 3**).

Second, we focus on the quartile of prompts with the lowest mean rewards, which corresponds to the most difficult quartile. We follow the preference pair construction strategy in Meng et al. (2024); Li & Khashabi (2025); Xiao et al. (2025), which selects the response of the lowest reward as the rejected and selects the response of the highest reward as the chosen from $n$ samples. We observe that this quartile of prompts tends to yield inferior performance than an equal number of prompts from the remaining set when training through DPO. In addition, this quartile of prompts will not lead to performance improvement when mixed with the remaining prompts for training. Our results further indicate that prompt difficulty correlates with model capacity. A sufficiently capable LLM can close the performance gap between difficult and easy prompts (**Section 4**).

Third, we propose three approaches to improve the final self-play preference optimization performance of models by mitigating the hard prompt issue: (1) curriculum learning (Bengio et al., 2009) that progressively trains from easy to hard prompts; (2) improving the quality of the chosen response for difficult prompts; and (3) removing the most difficult $k$ percent of prompts. We find that pruning difficult prompts is simple yet effective, whereas training from easy to hard prompts and improving chosen responses do not translate into final performance gains in our setting (**Section 5**).

To conclude, we highlight the overlooked role of prompts in self-play preference optimization in this work. We establish mean reward as a practical proxy for prompt difficulty, and show that difficult prompts contribute little to alignment. We show the performance difference between hard and easy prompts when optimizing policy models with DPO. We also share our attempts to mitigate this issue, including the unsuccessful trials. We encourage future research to revisit the design and utilization of prompts in alignment pipelines, ensuring that self-play preference optimization fully leverages the spectrum of prompt difficulty rather than being hindered by it.

## 2 PRELIMINARIES

### 2.1 DIRECT PREFERENCE OPTIMIZATION

Different from RLHF, which compresses human preferences into a reward model, DPO (Meng et al., 2024) directly aligns language models with human preferences. DPO is one of the most widely used methods for preference optimization, which reformulates the reward function $r$ into a closed-form expression aligned with the optimal policy model.

$$r(x, y) = \beta \log \frac{\pi_\theta(y|x)}{\pi_{\text{ref}}(y|x)} + \beta \log Z(x)$$

where $\pi_\theta$ denotes the policy model, $\pi_{\text{ref}}$ represents the reference model (usually the supervised fine-tuned checkpoint), and $Z(x)$ is the partition function. By embedding this reward formulation into the Bradley-Terry (BT) ranking framework (Bradley & Terry, 1952), the probability of preference $p(y_w > y_l|x)$ is calculated as $\sigma(r(x, y_w) - r(x, y_l))$, where $\sigma$ is the sigmoid function. Accordingly, DPO circumvents reliance on a reward model by directly leveraging the policy model, which yields the following objective.

$$\mathcal{L}_{\text{DPO}}(\pi_\theta; \pi_{\text{ref}}) =$$
$$- \mathbb{E}_{(x,y_w,y_l)\sim\mathcal{D}}\Big[ \log \sigma(r(x, y_w) - r(x, y_l)) \Big]$$

where $r(x, y) = \beta \log \frac{\pi_\theta(y|x)}{\pi_{\text{ref}}(y|x)}$.

## 2.2 PREFERENCE PAIR CONSTRUCTION

Given an LLM, a reward function, and a pool of prompts, multiple candidate responses can be sampled for each prompt from language models. The reward function (e.g., a reward model) is then used to score these responses. In general, responses that receive higher reward scores tend to correspond to higher-quality outputs.

For each prompt, the response with the highest reward is selected as the chosen response, while the response with the lowest reward is selected as the rejected response to form a preference pair. Previous work has shown that using as few as 5 samples per prompt is sufficient to achieve significant performance gains (Meng et al., 2024; Li & Khashabi, 2025). In this work, we adopt this pipeline to construct preference pairs of prompts for DPO.

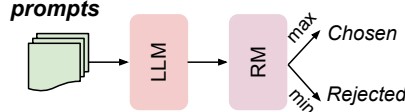

Figure 1: We follow this preference pair construction approach introduced by Meng et al. (2024); Li & Khashabi (2025). The samples of maximal reward and minimal reward can make a preference pair for training.

## 3 MEAN SAMPLE REWARD AS A PROXY OF PROMPT DIFFICULTY

The first question in our study of self-play preference optimization is how to quantify the difficulty of a prompt for an LLM. In this section, we elaborate on how to measure prompt difficulty based on the mean reward of multiple sampled responses. In addition, we statistically demonstrate that our difficulty ranking is transferable to some extent between LLMs (RMs).

### 3.1 DEFINITION OF PROMPT DIFFICULTY

Given a set of questions $\mathcal{D} = \{P_i\}_{i=1}^k$, we sample $N$ candidate responses per prompt from the policy model $\pi_\theta$ and score them with a reward model $r$. The reward of $N$ candidate samples of the $i$-th prompt is $\{r_{ij}\}_{j=1}^N$. The mean of these $N$ reward values is then used as a proxy for the difficulty of the prompt:

$$D(P_i) = \frac{1}{N} \sum_{j=1}^N r_{ij}.$$

Our intuition is straightforward: prompts with lower mean rewards are generally harder for LLMs, since they consistently elicit lower-quality responses across multiple samples, whereas prompts with higher mean rewards are easier. This allows us to rank or partition prompts into subsets according to their estimated difficulty score.

**Experimental Details.** For policy models, we employ LLAMA-3.1-TULU-3-8B-SFT (Lambert et al., 2024) and MISTRAL-7B-INSTRUCT-V0.2 [2]. To compute rewards, we leverage the publicly

---

[2] For brevity, we may refer to them as Tulu and Mistral in the rest of this paper.

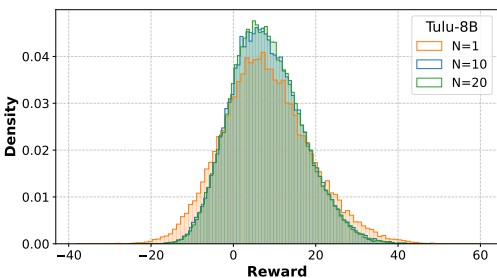 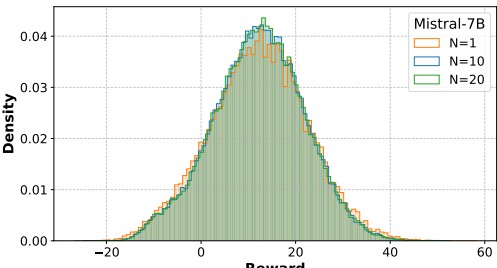

Figure 2: We show the mean reward distribution of $N$ sampled responses per prompt on LLAMA-3.1-TULU-3-8B-SFT and MISTRAL-7B-INSTRUCT-V0.2 for prompts of UltraFeedback (Cui et al., 2024). We find 10 samples per prompt are sufficient to obtain a stable estimate.

available reward model SKYWORK-REWARD-LLAMA-3.1-8B-V0.2 (Liu et al., 2024a). Our implementation is based on vLLM (Kwon et al., 2023) for efficient inference with a temperature of $0.8$ and a maximal generation length of $2048$. Our prompts are from UltraFeedback (Cui et al., 2024), which covers more than 61K prompts of high quality from diverse domains.

**Observation.** As shown in Figure 2, we present the mean reward distribution of 1, 10, and 20 samples per prompt from UltraFeedback (Cui et al., 2024). Generally, increasing the sampling budget produces more stable and consistent estimates of relative prompt difficulty. In our experiment, we observe that the mean reward of 10 samples per prompt can provide a stable estimate, which is very close to the distribution of 20 samples per prompt [3]. Therefore, we use the mean reward of 10 samples per prompt as a proxy for question difficulty throughout this work.

### 3.2 TRANSFERABILITY ACROSS MODELS

An additional question is whether prompt difficulty, measured by mean reward, is model-specific or generalizable. To investigate this, we compare the difficulty rank of the whole prompt set for Tulu and Llama. We find that the Spearman score between Tulu and Llama is about $0.68$. And there are more than 9,000 common prompts in the most difficult quartile of prompts for Tulu and Llama. Our results show that prompts identified as difficult for one model tend to remain difficult for another one, suggesting that relative prompt difficulty is not strictly tied to a single policy model but is transferable across LLMs to some extent. More exploration of transferability between reward models can be found in Appendix A.

### 4 IMPACT OF DIFFICULT PROMPTS ON SELF-PLAY PREFERENCE LEARNING

In this section, we mainly study the hardest quartile (bottom $25\%$) of prompts. We observe that, given the same number of prompts, hard prompts underperform simpler ones. Additionally, excluding this quartile of prompts from the full set results in slight but consistent gains in overall performance. In the end, we point out that the performance gap between this hard quartile and easier prompts will close if policy models are sufficiently capable.

**Experimental Details.** We first sort prompts in UltraFeedback (Cui et al., 2024) by the difficulty metrics introduced in Section 3. We use LLAMA-3.1-TULU-3-8B-SFT and MISTRAL-7B-INSTRUCT-V0.2 as policy models to sample responses for preference pair construction and further train with DPO (Rafailov et al., 2023). We employ the publicly available reward model SKYWORK-REWARD-LLAMA-3.1-8B-V0.2 to score responses. We follow the strategy in Section 2.2 to construct preference pairs by sampling 5 responses per prompt for DPO. We evaluate model performance on AlpacaEval 2 (Dubois et al., 2023; 2024), which is the most widely used benchmark in this field. AlpacaEval 2 consists of 805 questions from multiple domains and tasks, which enables

---

[3]We also have a Kolmogorov–Smirnov (KS) test between the reward distribution of 10 and 20 samples per prompt. It turns out that the p-value is lower than 0.05, which further supports that 10 samples per prompt is sufficient here.

Table 1: The quartile of hardest prompts (bottom 25%) underperforms a equal number of easier prompts sampled form top 75% on AlpacaEval 2. Last row denotes results averaged over three runs for easier prompts.

| Method | Llama-3.1-Tulu-3-8B | | | Mistral-7B-Instruct-v0.2 | | |
|---|---|---|---|---|---|---|
| | LC (%) | WR (%) | Length | LC (%) | WR (%) | Length |
| Original Model | 18.10 | 10.90 | 1115 | 17.63 | 14.68 | 1594 |
| Hard Prompts | 24.23 | 18.78 | 1502 | 25.96 | 23.11 | 1718 |
| Easier Prompts | | | | | | |
| Run 1 | 29.61 | 31.93 | 2101 | 28.81 | 29.07 | 2044 |
| Run 2 | 28.15 | 30.99 | 2115 | 28.32 | 28.53 | 2032 |
| Run 3 | 30.83 | 33.35 | 2121 | 28.73 | 27.32 | 2017 |
| Average (Easier Prompts) | 29.53 | 32.09 | 2112 | 28.62 | 28.31 | 2031 |

a comprehensive assessment of LLMs. Both length-controlled win rate and vanilla win rate [4] are reported. . The decoding temperature is 0.9 and 0.7 for Tulu and Mistral during evaluation, respectively. More details about training can be found in Appendix B.

### 4.1 HARDER PROMPTS UNDERPERFORM EASIER PROMPTS

We examine the hardest quartile (bottom 25%) of prompts and compare it to subsets drawn from the remaining easier set. Specifically, we sample an equal number of prompts from the remaining set with three different seeds to ensure a fair and robust evaluation. This approach reduces the variance introduced by random selection and provides a more reliable evaluation. We also aggregate the results by averaging over the three runs, which allows us to better capture the performance difference between the hardest quartile and other easier prompts. We show their result training through DPO in Table 1. Across both backbone models (Tulu and Mistral) and multiple random realizations, performance on the most difficult 25% of prompts exhibits lower performance than that of random subsets from the remaining pool in both length-controlled win rate and vanilla win rate. These results highlight an underlying weakness of self-play preference optimization when applied to hard prompts.

> **Takeaway.** The hardest quartile of prompts produces smaller performance gains, limiting effective optimization under DPO. Prioritizing easier items can yield clearer guidance and greater benefits for optimizing self-play preferences.

### 4.2 HARD PROMPTS MAY HURT FINAL PERFORMANCE

To further examine the impact of the most difficult quartile of prompts on final performance, we train models with and without this subset under DPO. In the case without this quartile of prompts, the performance of two backbone models (Tulu and Mistral) improves on AlpacaEval 2 (Figure 3) despite discarding 25% of the training data, which also saves 25% of the computing costs. We can observe a slight but consistent improvement in terms of vanilla win rate and length-controlled win rate, indicating the gains are not attributable to longer responses or verbosity effects. In practice, this means that naive scaling of self-play data without accounting for prompt difficulty may yield diminishing or even negative returns. It highlights the trade-off between prompt quantity and prompt difficulty, which is of practical significance.

> **Takeaway.** Incorporating all prompts into training may not improve the overall performance of LLMs. By contrast, dropping the hardest quartile of prompts achieves slight but consistent performance gains and saves compute.

---

[4]For brevity, we refer to length-controlled win rate as LC and refer to win rate as WR in most tables and figures of this paper

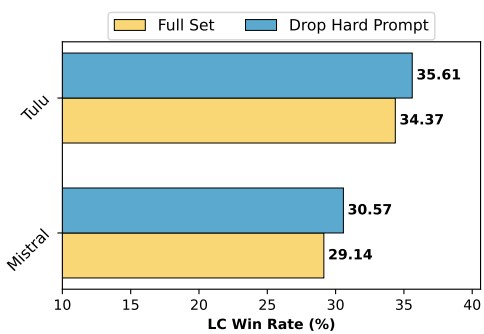 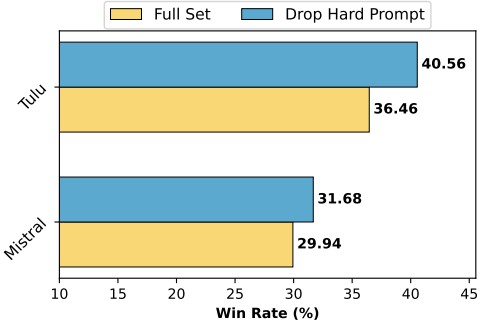

Figure 3: We present the results of dropping the most difficult quartile of prompts and the full set results on AlpacaEval 2. We can see that incorporating the hardest quartile of prompts into training may hurt the final performance of models.

### 4.3 THE GAP MAY CLOSE IF LLMS ARE STRONG ENOUGH

The impact of prompt difficulty is not uniform across LLMs of varying capacity. Although weaker policy models such as TULU-3-8B-SFT suffer from a clear performance gap between hard and easier prompts (Section 4.1), this trend does not hold for a stronger model such as LLAMA-3.1-8B-INSTRUCT. Table 2 shows that LLAMA-3.1-8B-INSTRUCT, a more capable model, achieves comparable performance on hard and easy prompts, with almost no gap in length-controlled win rate and vanilla win rate.

This finding suggests that the interplay between model capacity and prompt complexity influences the effect of challenging prompts. For weaker models, hard prompts tend to underperform easier prompts when performing self-play preference optimization. In contrast, stronger models are more robust to prompts of varying difficulty.

Table 2: We find that LLAMA-3.1-8B-INSTRUCT exhibits no significant performance gap between hard prompts and easier prompts, which suggests that increased model capacity can make model more tolerant to hard prompts to colse the performance gap.

| Method | LC(%) | WR(%) | Len. |
|---|---|---|---|
| Original Model | 23.75 | 24.23 | 1972 |
| Hard Prompts | 34.31 | 38.32 | 2059 |
| Easier Prompts | | | |
| Run 1 | 34.59 | 38.32 | 2212 |
| Run 2 | 34.45 | 38.62 | 2223 |
| Run 3 | 35.25 | 39.19 | 2230 |
| Average (Easier) | 34.76 | 38.71 | 2221 |

Although sufficiently capable models may bridge the performance gap on hard prompts of UltraFeedback Cui et al. (2024) in our experiment, the existence of hard prompts in the real world remains unavoidable. This underscores the practical importance of our work, which also motivates us to study prompts.

> **Takeaway.** The performance gap between difficult prompts and easy prompts can close when the capacity of LLMs is strong enough. It indicates that prompt difficulty is a relative notion—what is hard for a weaker model may become manageable for a stronger one.

## 5 HOW TO MITIGATE THE DIFFICULT PROMPT ISSUE

Our findings suggest that difficult prompts, as identified by their low mean sample rewards, tend to underperform in self-play preference optimization and can even slightly degrade overall performance. In this section, we investigate three strategies to mitigate the adverse impact of difficult prompts while also documenting unsuccessful attempts to provide a comprehensive account of our study. We adopt the experimental settings from Section 4, unless otherwise stated.

## 5.1 Unsuccessful Attempts And Implications

**Training from Easy to Hard Prompts.** Inspired by curriculum learning (Bengio et al., 2009; Graves et al., 2017), we propose to train models progressively, starting with easy prompts and gradually incorporating more difficult ones. The difficulty score of the prompts, based on the mean reward of sampled responses, serves as a natural foundation for constructing such a curriculum. This setup enables us to assess whether curriculum learning can enhance the standard approach of training on the full prompt set in random order.

As shown in Table 3, we find that this curriculum learning paradigm leads to no improvement in the performance of the full set of random order on the Tulu model.

**Constructing Better Preference Pairs.** One hypothesis for the limited contribution of hard prompts is that their chosen responses are more likely to exhibit lower quality, which may hinder self-play preference optimization. And a language model may struggle to generate high-quality responses within a budget of only 5 samples per prompt. Thus, we attempt to generate better chosen responses for the most difficult $k$ percent of prompts by increasing the sample budget to 20 samples per prompt (Xiao et al., 2025). By selecting the completion of the maximal reward score in 20 samples as the chosen response for the most difficult $k$ percent of prompts, we construct less noisy preference pairs for DPO. In addition, we also tried to sample chosen responses for the most difficult $k$ percent of prompts from a more capable LLM, LLAMA-3-70B-INSTRUCT.

Table 3: We present our attempts that fail to improve the performance on AlpacaEval 2. They are (1) training from easy to hard prompts, (2) increasing the number of samples, (3) sampling chosen responses from more capable models, respectively.

| Method | LC (%) | WR (%) | Len. |
|---|---|---|---|
| Full Set | 34.37 | 36.46 | 2101 |
| Easy→Hard | 33.78 | 34.84 | 2093 |
| Chosen in 20 Samples | | | |
| $k=20(\%)$ | 33.64 | 36.09 | 2129 |
| $k=40(\%)$ | 33.17 | 34.47 | 2071 |
| Chosen from Llama-3-70B | | | |
| $k=20(\%)$ | 34.41 | 38.05 | 2238 |
| $k=40(\%)$ | 26.08 | 32.96 | 2659 |

The results of these two strategies can also be found in Table 3. Both approaches do not yield better performance compared to the results of the full set, whose sample budget is 5 per prompt. These findings rule out the possibility that the hard prompt issue arises from low-quality chosen responses, and instead indicate that the capacity of language models fundamentally constrains it.

> **Implication.** Replacing the chosen responses of hard prompts with higher-quality samples does not improve self-play preference optimization, reinforcing our view that the impact of difficult prompts may be intrinsically constrained by model capacity.

## 5.2 A Simple Solution: Remove the Most Difficult Prompts

Based on our findings in Section 4, which shows that dropping the hardest quartile improves final performance, we examine an adaptive pruning strategy in this section. Specifically, we first rank prompts by their difficulty scores and remove the most difficult $k$ percent of prompts before constructing preference pairs. The value of $k$, which controls the fraction of prompts removed, depends on the prompt difficulty and the capacity of the models. In practice, $k$ can be tuned with a benchmark such as AlpacaEval 2.

**Experimental Details.** We mainly follow the experimental setting described in Section 4. We remove 30 ($k=30$) percent and 50 ($k=50$) percent of the most difficult prompts for Tulu and Mistral, respectively. We evaluate model performance on **AlpacaEval 2** (Dubois et al., 2023; 2024) and **Arena-Hard v0.1** (Li et al., 2024a). More details can be found in Appendix B.

In addition to the full set of results, we also include the results of dropping 30 percent of prompts for Tulu and 50 percent for Mistral randomly, which are named *Random*.

**Results.** As shown in Table 4, our method outperforms the complete set and random drop results on AlpacaEval 2 and Arena-Hard, with significantly less training compute. The results support our

Table 4: We report evaluation results on ALPACAEVAL 2 and ARENA-HARD V0.1. For our method, we remove 30% and 50% most difficult prompts on Llama-3.1-Tulu-3-8B and Mistral-7B-Instruct-v0.2 respectively. *Random* means that we remove a equal number of prompts randomly from *full set* as our method.

| Method | Llama-3.1-Tulu-3-8B | | | Mistral-7B-Instruct-v0.2 | | |
| | AlpacaEval 2 | | Arena-Hard | AlpacaEval 2 | | Arena-Hard |
| | LC (%) | WR (%) | WR (%) | LC (%) | WR (%) | WR (%) |
|---|---|---|---|---|---|---|
| Full Set | 34.37 | 36.46 | 39.2 | 29.24 | 29.94 | 20.1 |
| Random | 33.53 | 33.48 | 36.2 | 29.31 | 30.93 | 19.3 |
| Ours | **36.87** | **38.20** | **39.8** | **30.99** | **32.42** | **20.6** |

hypothesis that difficult prompts do not help the preference optimization process, limiting effective learning. In contrast, removing a controlled portion of the hardest prompts not only improves performance, but also reduces training cost. This suggests that pruning strategies based on prompt difficulty can serve as a simple yet effective approach to enhance the efficiency of self-play preference optimization pipelines. The results of more reward models can be found in Appendix C.

**Remove Varying Portion of Hard Prompts.** To further investigate the sensitivity of self-play optimization to the proportion of difficult prompts removed, we vary $k$ from 10 to 50 and report the results in Figure 4. We find that the model performance increases steadily as $k$ grows from 10 to 30, with the peak performance observed at $k$=30. Beyond this point, removing a larger fraction of prompts begins to degrade performance, as valuable training signals are discarded together with the hardest prompts. This trend suggests that moderate pruning of difficult prompts is beneficial, while overly aggressive pruning risks underutilizing the training corpus. In general, these results reinforce the conclusion that prompt difficulty serves as an effective criterion for adaptive pruning in preference optimization. More experiments can be found in Appendix D.

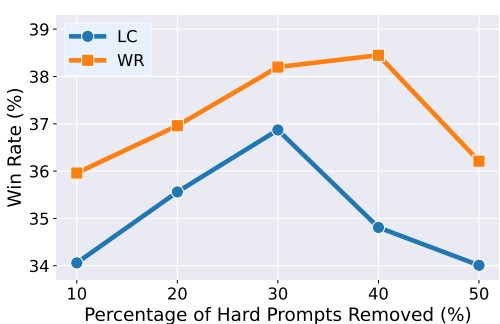

Figure 4: We present the performance of model on AlpacaEval 2 as we change $k$ from 10 to 50 percent on Tulu. Performance first improves and then degrades as we prune more and more hard prompts. Performance reaches its peak when we remove about 30 percent of difficult prompts.

## 6 RELATED WORK

**Training Sample Selection.** Training sample selection is a long–standing lever for both generalization and computational efficiency (Kumar et al., 2010). For instance, LIMA(Zhou et al., 2023) shows that a carefully curated small instruction set can deliver surprisingly effective alignment, reinforcing a "less is more" perspective (Muennighoff et al., 2025). In preference learning, sample selection has operated chiefly at the *response* level, e.g., RAFT-style (Dong et al., 2023) filtering and statistical rejection that keep high-quality chosen responses and prune noisy pairs, or simple mixing rules for off- vs. on-policy data. Still, recent work sharpens the principle to account for *example difficulty vs. model capacity*. Specifically, Gao et al. (2025) shows that overly difficult preference examples can hinder alignment and proposes Xiao et al. (2025), which filters such items and yields notable gains in instruction following (Dubois et al., 2023). Unlike previous work that assumes a fixed set of prompts and aims to construct more effective preference pairs, we instead score the prompts by the mean reward and investigate how the prompts of varying difficulty influence preference optimization, especially the hard ones.

**Reinforcement Learning from Human Feedback.** RLHF is a leading paradigm for aligning large language models with human preferences in natural language generation (Ouyang et al., 2022; Touvron et al., 2023). Nevertheless, it can suffer from training instability and operational complexity inherent to reinforcement learning, as well as its multistage design, which may introduce biases and encourage verbose outputs. DPO (Rafailov et al., 2023) and its variants (Meng et al., 2024; Ethayarajh et al., 2024; Han et al., 2024) were proposed to mitigate these issues by directly fitting the policy model to pairwise preference data, thereby removing the explicit reward-modeling phase and simplifying optimization. As more capable reward models have become publicly available (Jiang et al., 2023; Wang et al., 2024b;a; Liu et al., 2024b), a common practice (Dong et al., 2023; Liu et al., 2024c; Meng et al., 2024; Li & Khashabi, 2025) is to use them to score and select self-generated samples, enabling DPO-based training. Unlike prior work that focuses on reward models or preference pair construction, we shift selection to *prompt level*: we operationalize prompt difficulty via the mean reward score over $N$ samples, show a capacity-dependent hard prompt penalty, and demonstrate that pruning the hardest prompts improves self-play performance.

**Synthetic Data for LLMs.** While human–curated corpora continue to serve as a benchmark resource for advancing natural language processing (NLP) systems (Bai et al., 2022; Köpf et al., 2023), their large–scale expansion remains constrained by substantial financial and labor costs. As a cost–effective alternative, the use of synthetic data has received increasing attention (West et al., 2022; Hsieh et al., 2023; Wang et al., 2023; Dong et al., 2024; Li et al., 2024b; Kim et al., 2025), particularly through the use of advanced large language models (LLMs) to automatically generate high–quality training corpora (Tajwar et al., 2024; Dong et al., 2023; Agarwal et al., 2024; Chen et al., 2025). Within this research trajectory, self–generated on-policy data has emerged as both a practical and computationally efficient paradigm, and has therefore attracted considerable scholarly interest. The present work is situated within this line of work, employing synthetic data to construct preference training pairs aimed at enhancing model performance.

# 7 CONCLUSION

In this work, we investigate the often-overlooked role of prompts in self-play preference optimization. We introduce the mean reward of multiple sampled responses as a practical proxy for prompt difficulty, showing that difficult prompts consistently underperform easier ones in self-play preference optimization. Through systematic analysis, we demonstrate that incorporating these difficult prompts does not improve performance but slightly degrades it. We further find that stronger models can almost close the gap, underscoring the interaction between model capacity and prompt complexity. We also explore several strategies to mitigate the hard prompt issue. Although curriculum training and the construction of higher-quality preference pairs failed to yield improvements, a simple pruning strategy of removing a controlled fraction of the hardest prompts proved effective. In general, our findings suggest that prompt difficulty deserves careful consideration in preference optimization. Future research should extend this line of inquiry by developing adaptive methods that dynamically weight or filter prompts based on model capacity, ensuring that self-play preference optimization can fully leverage available data without being hindered by inherently difficult prompts.

ETHICS STATEMENT

We did not collect new human-subject data and processed no personally identifiable information. We assessed and observed no potential harms (e.g., bias amplification, toxic outputs, and dual-use risks) in our work.

REPRODUCIBILITY STATEMENT

We are committed to ensuring the reproducibility of our work. Details about data generated and used for preference optimization are provided in our paper. Upon publication, we plan to release the source code and data to the public. Detailed descriptions of our experimental setup and hyperparameters are also provided throughout the paper.

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

## A  TRANSFERABILITY BETWEEN REWARD MODELS

We also explore the difficulty transferability between different reward models, SKYWORK-REWARD-LLAMA-3.1-8B-V0.2 and ARMORM-LLAMA3-8B-V0.1 (Wang et al., 2024b). They are training with different loss design and data. We find that the difficulty scores between them computed with 10 samples per prompt have a Spearman score about 0.72. In addition, there are more than 9.4k common prompts in the most difficult quartile between them. This result shows that prompts identified as difficult for a reward model tend to remain difficult for another one, suggesting that prompt difficulty is transferable across reward models to some extent.

## B  IMPLEMENTATION DETAILS

**Training Hyperparameter.** For all of our experiments, we use Trl (von Werra et al., 2020) for training. Initially, we performed hyperparameter sweeps for $\beta = \{0.01, 0.1, 0.5\}$ and max learning rate $= \{3e-7, 5e-6, 1e-6\}$ as initial exploration. We report the result on $\beta = 0.01$ and max learning rate $= 3e-7$ for experiments on LLAMA-3.1-TULU-3-8B-SFT. We report the result on $\beta = 0.01$ and max learning rate $= 3e-7$ for experiments on MISTRAL-7B-INSTRUCT-V0.2. We employ 10% of steps as training warmup.

**Evaluation Hyperparameter.** We maintain a maximal generation length, 2048, for both models. The temperature for MISTRAL-7B-INSTRUCT-V0.2 is 0.7 and is 0.9 for LLAMA-3.1-TULU-3-8B-SFT when evaluating AlpacaEval 2. For Arena-Hard, the maximal length is 4098 for both models and we use default temperature 0.

## C  RESULTS ON MORE REWARD MODELS

In this part, we demonstrate the effectiveness of removing the most difficult $k$ percent of prompts for LLAMA-3.1-TULU-3-8B-SFT on more reward models, ARMORM-LLAMA3-8B-V0.1 (Wang et al., 2024b) and LLAMA-3.1-8B-INSTRUCT-RM-RB2 (Malik et al., 2025). Specifically, we remove the most difficult 30% of prompt and evaluate the model after DPO training with AlpacaEval 2. The results are shown in Table 5. Our conclusions hold across different reward models.

| Method | ArmoRM-Llama3-8B-v0.1 | | |
|---|---|---|---|
| | LC | WR | Length |
| full | 36.39 | 39.13 | 2313 |
| random | 37.29 | 35.22 | 1966 |
| our | **39.01** | **39.94** | 1935 |
| Method | Llama-3.1-8B-Instruct-RM-RB2 | | |
| | LC | WR | Length |
| full | 37.59 | 37.83 | 2045 |
| random | 36.60 | 36.46 | 2021 |
| our | **39.89** | **40.68** | 2074 |

Table 5: We demonstrate the validity of removing hard prompts on more reward models, ARMORM-LLAMA3-8B-V0.1 and LLAMA-3.1-8B-INSTRUCT-RM-RB2.

## D  REMOVE VARYING PORTION OF HARD PROMPTS FOR MISTRAL

We vary $k$ from 10% to 60% and report the results in Figure 5. There is an increasing trend as $k$ grows from 10% to 60%, with the peak performance observed at $k = 50\%$. Beyond this point, removing a larger fraction of prompts begins to degrade performance.

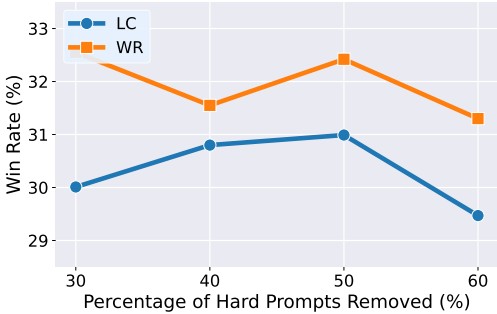

Figure 5: We present the performance of model on AlpacaEval 2 as we change $k$ from 10 to 50 percent on Mistral. Performance first improve and then degrade as we prune more and more hard prompts. Performance reaches peak when we remove about 50 percent of most difficult prompts.

