# OpenReview forum: "On the Role of Difficult Prompts in Self-Play Preference Optimization"
_ICLR.cc/2026/Conference — ICLR 2026 Conference Withdrawn Submission_

### Official Review · Reviewer_voFf · 2025-10-28

**Soundness:** 2
**Presentation:** 2
**Contribution:** 2
**Rating:** 4
**Confidence:** 4

**Summary:**

This paper investigates how prompt difficulty, defined by the mean reward of model-generated responses, affects self-play preference optimization (SPPO) with DPO. The authors find that difficult prompts provide weaker learning signals and that pruning the hardest 25–30 percent improves both performance and efficiency.

**Strengths:**

1. The paper is easy to follow. The motivation is clear, and the writing is well organized.

2. Experiments are carefully designed and reproduce consistent trends across two backbone models (Tulu-3-8B and Mistral-7B).

3. From the motivation perspective, the authors conduct a series of ablation studies and summarize several informative findings about how prompt difficulty affects data efficiency in self-play preference optimization.

**Weaknesses:**

1. The overall novelty is moderate, and the scope of the conclusions is limited. Most results are empirical observations without strong theoretical analysis.
2. The generalization of conclusions is unclear. All analysis and experiments are conducted under DPO. It would be valuable to discuss whether similar behaviors occur in other alignment frameworks such as SimPO, KTO.
3. The task-domain generalization is uncertain. The study seems restricted to instruction-following prompts. For more complex domains such as coding or mathematical reasoning, simply pruning the hardest prompts may harm performance or generalization.
4. Regarding the “response improvement” strategy, the authors report limited benefits. However, prior work such as Filtered DPO (Morimura et al., 2024) shows that filtering out low-quality chosen samples that underperform the model’s own responses can improve alignment quality. It would be helpful to clarify whether the limited effect observed here is due to the conceptual difference between improving and filtering chosen responses, or to specific factors in the self-play DPO setting.

**Questions:**

Please see the “Weaknesses” section above.

---

### Official Review · Reviewer_DTwW · 2025-10-31

**Soundness:** 4
**Presentation:** 4
**Contribution:** 1
**Rating:** 2
**Confidence:** 4

**Summary:**

This paper studies how prompt difficulty affects self-play preference optimization in large language models. Using the mean reward of sampled responses as a proxy for difficulty, the authors find that difficult prompts lead to worse optimization performance compared to easy ones, and including them in training slightly degrades overall performance. The performance gap narrows with larger model capacity, suggesting that difficulty interacts with model scale. The paper further explores strategies to mitigate the negative effects of hard prompts, showing that selective removal of challenging prompts can improve overall self-play outcomes.

**Strengths:**

- This paper introduces a practical and transferable metric using the mean reward of sampled responses as a proxy for prompt difficulty, validated across different reward models and datasets.
- This paper proposes and evaluates three practical mitigation strategies—curriculum learning, response quality improvement, and pruning difficult prompts—to address the “hard prompt” issue.
- The writing is easy to follow, and the claims of the paper are adequately supported with experiments.

**Weaknesses:**

- The proposed “difficulty issue” seems hard to scale up. Since difficult data have almost no impact on the LLaMA-8B model, this problem may not exist for larger and more powerful modern models.

- The motivation for studying data difficulty lacks sufficient novelty, as this topic has already been widely investigated in many existing works.

- The proposed solutions for addressing the difficulty problem are straightforward and lack originality. For example, using the average score to measure data difficulty and filtering out difficult samples are simple approaches that provide limited contribution to advancing self-play DPO research.

**Questions:**

What are the implementation details of the curriculum learning paradigm used in this paper? How do the authors adjust the difficulty of the data during training?

---

### Official Review · Reviewer_2H8b · 2025-10-31

**Soundness:** 3
**Presentation:** 2
**Contribution:** 2
**Rating:** 2
**Confidence:** 3

**Summary:**

The paper investigates the impact of prompt difficulty on self-play preference optimization, a popular technique for aligning LLMs. It proposes a metric for prompt difficulty based on the mean reward of N sampled responses from a policy model.
The core findings are:
- Difficult prompts lead to inferior performance in Direct Preference Optimization (DPO) compared to easier prompts.
- Including these difficult prompts in the training set can slightly degrade overall model performance.
- This performance gap between easy and hard prompts diminishes as model capacity increases.
Based on these findings, the authors show that a simple strategy of pruning the most difficult k% of prompts improves final model performance and reduces computational cost. The paper also transparently reports on unsuccessful mitigation strategies, such as curriculum learning.

**Strengths:**

- The paper is well-written and has a logical flow that is easy to follow.
- The paper's main takeaway—that pruning difficult prompts is an effective strategy—is simple to understand and implement, making it highly practical for researchers and engineers working on LLM alignment.
- Sharing negative results demonstrates scientific rigor and provides the community with valuable information.

**Weaknesses:**

- The entire framework hinges on the assumption that the reward model (RM) is a reliable and unbiased judge of quality. The "difficulty" metric may inadvertently be capturing the RM's deficiencies. Even a small-scale human study would signficiantly boost the reliability of the findings.
- Constructing Better Preference Pairs seems somewhat shallow, as the strategies used do not necessarily provide good pairs for challenging benchmarks such as AlpacaEval 2.0. Llama 3 8B instruct and Llama 70B achieve 22.9% and 34.4% lc win rate, respectively and share a lot of failure cases, while the the sota models achieve 50% +. As such, is is possible that even after the selection, the pairs were not significantly better.
- The claims for the method's simplicity and practicality are somewhat undermined by the introduction of k parameter as it requires a hold-out evaluation benchmark and significant computation to tune.
- While nicely formulated, the difficulty metric used, as well as the pruning of the dataset based on it are fairly close to standard approaches, and do not provide a major change in the pipeline.
- There is a lack of qualitative study of samples, especially for if there is some qualities shared across the identified difficult samples and how excluding them might affect the usability of the model.

**Questions:**

- Any insights on the hard prompts? Is it possible that "hard" prompts are simply those on topics or in styles that the specific RM is biased against?
- Would you expect the trend for Better Preference Pairs to hold with stronger models that are better able to handle the benchmark?
- Would you expect the difficulty to be interpretable or correlated with human judgment?

---

### Official Review · Reviewer_NjuL · 2025-11-01

**Soundness:** 2
**Presentation:** 2
**Contribution:** 1
**Rating:** 2
**Confidence:** 5

**Summary:**

The paper discusses the role of prompts’ difficulty in direct preference optimization training. The expected reward of a prompt is proposed as a difficulty metric, and the experimental results show that removing difficult prompts from the dataset improves the alignment performance.

**Strengths:**

The paper addresses an important problem of selecting prompts for alignment.

**Weaknesses:**

- The discussion presented in this paper is mostly empirical and descriptive, without delving deeper into the root causes. The current manuscript does not contribute much to understanding **why** such difficult prompts are not helpful. A more in-depth analysis from diverse perspectives, for example, examining training dynamics, the gradient of the loss function, or even conducting qualitative analyses of the prompts and responses, would greatly improve the paper.
- The conclusion of the paper, which calls for removing the difficult prompts, is too naive and unsatisfactory.
- The use of the term *“self-play preference optimization”* is somewhat misleading. I would recommend replacing it with *“direct preference optimization.”* There exists a large body of work beyond DPO that is often referred to as *“self-play preference optimization,”* such as Nash-MD (https://arxiv.org/html/2312.00886v3) and SPPO (https://arxiv.org/abs/2405.00675). However, the current manuscript does not cover these methods and therefore has a limited scope.

**Questions:**

- If difficult prompts are unhelpful for learning, are easy prompts unhelpful as well?
- DPO is originally proposed to learn directly from the preference labels, without a reward model. Are the experiments and analysis in this paper applicable to the setting where there is no reward model? For example, how would you compute the difficulty of a prompt without having access to a reward model?

---

### Note · Authors · 2025-11-12

**Comment:**

I will come back better.

**Withdrawal Confirmation:**

I have read and agree with the venue's withdrawal policy on behalf of myself and my co-authors.